# Meta-trained agents implement Bayes-optimal agents

**Vladimir Mikulik,**[*] **Grégoire Delétang,**[*] **Tom McGrath,**[*] **Tim Genewein,**[*]
**Miljan Martic, Shane Legg, Pedro A. Ortega**[†]
DeepMind
London, UK

## Abstract

Memory-based meta-learning is a powerful technique to build agents that adapt fast to any task within a target distribution. A previous theoretical study has argued that this remarkable performance is because the meta-training protocol incentivises agents to behave Bayes-optimally. We empirically investigate this claim on a number of prediction and bandit tasks. Inspired by ideas from theoretical computer science, we show that meta-learned and Bayes-optimal agents not only behave alike, but they even share a similar computational structure, in the sense that one agent system can approximately simulate the other. Furthermore, we show that Bayes-optimal agents are fixed points of the meta-learning dynamics. Our results suggest that memory-based meta-learning might serve as a general technique for numerically approximating Bayes-optimal agents—that is, even for task distributions for which we currently don't possess tractable models.

## 1 Introduction

Within the paradigm of learning-to-learn, memory-based meta-learning is a powerful technique to create agents that adapt fast to any task drawn from a target distribution [1–6]. In addition, it has been claimed that meta-learning might be a key tool for creating systems that generalize to unseen environments [7]. This claim is also partly supported by studies in computational neuroscience, where experimental studies with human subjects have shown that fast skill adaptation relies on task variation [8, 9]. Due to this, understanding how meta-learned agents acquire their representational structure and perform their computations is of paramount importance, as it can inform architectural choices, design of training tasks, and address questions about generalisation and safety in artificial intelligence.

Previous theoretical work has argued that agents that fully optimise a meta-learning objective are Bayes-optimal by construction, because meta-learning objectives are Monte-Carlo approximations of Bayes-optimality objectives [10]. This is striking, as Bayes-optimal agents maximise returns (or minimise loss) by optimally trading off exploration versus exploitation [11]. The theory also makes a stronger, structural claim: namely, that meta-trained agents perform Bayesian updates "under the hood", where the computations are implemented via a state machine embedded in the memory dynamics that tracks the sufficient statistics of the uncertainties necessary for solving the task class.

Here we set out to empirically investigate the computational structure of meta-learned agents. However, this comes with non-trivial challenges. Artificial neural networks are infamous for their hard-to-interpret computational structure: they achieve remarkable performance on challenging tasks, but the computations underlying that performance remain elusive. Thus, while much work in explainable machine learning focuses on the I/O behaviour or memory content, only few investigate the internal dynamics that give rise to them through careful bespoke analysis—see e.g. [12–18].

---

[*]Equal contribution

[†]Correspondence to `{vmikulik|gdelt|mcgrathtom|timgen|pedroortega}@google.com`

To tackle these challenges, we adapt a relation from theoretical computer science to machine learning systems. Specifically, to compare agents at their computational level [19], we verify whether they can *approximately simulate* each other. The *quality* of the simulation can then be assessed in terms of both state and output similarity between the original and the simulation.

Thus, our main contribution is the investigation of the computational structure of RNN-based meta-learned solutions. Specifically, we compare the computations of meta-learned agents against the computations of Bayes-optimal agents in terms of their behaviour and internal representations on a set of prediction and reinforcement learning tasks with known optimal solutions. We show that on these tasks:

- Meta-learned agents *behave* like Bayes-optimal agents (Section 4.1). That is, the predictions and actions made by meta-learned agents are virtually indistinguishable from those of Bayes-optimal agents.

- During the course of meta-training, meta-learners *converge* to the Bayes-optimal solution (Section 4.2). We empirically show that Bayes-optimal policies are the fixed points of the learning dynamics.

- Meta-learned agents *represent* tasks like Bayes-optimal agents (Section 4.3). Specifically, the computational structures correspond to state machines embedded in (Euclidean) memory space, where the states encode the sufficient statistics of the task and produce optimal actions. We can approximately simulate computations performed by meta-learned agents with computations performed by Bayes-optimal agents.

## 2  Preliminaries

**Memory-based meta-learning**  Memory-based meta-learners are agents with memory that are trained on batches of finite-length roll-outs, where each roll-out is performed on a task drawn from a distribution. The emphasis on memory is crucial, as training then performs a search in algorithm space to find a suitable adaptive policy [20]. The agent is often implemented as a neural network with recurrent connections, like an RNN, most often using LSTMs [4, 21], or GRUs [22]. Such a network computes two functions $f_w$ and $g_w$ using weights $w \in \mathcal{W}$,

$$
\begin{aligned}
y_t &= f_w(x_t, s_{t-1}) &&\text{(output function)} \\
s_t &= g_w(x_t, s_{t-1}), &&\text{(state-transition function)}
\end{aligned}
\tag{1}
$$

that map the current input and previous state pair $(x_t, s_{t-1}) \in \mathcal{X} \times \mathcal{S}$ into the output $y_t \in \mathcal{Y}$ and the next state $s_t \in \mathcal{S}$ respectively. Here, $\mathcal{X}, \mathcal{Y}, \mathcal{S}$, and $\mathcal{W}$ are all vector spaces over the reals $\mathbb{R}$. An input $x_t \in \mathcal{X}$ encodes the instantaneous experience at time $t$, such as e.g. the last observation, action, and feedback signal; and an output $y_t \in \mathcal{Y}$ contains e.g. the logits for the current prediction or action probabilities. RNN meta-learners are typically trained using backpropagation through time (BPTT) [23, 24]. For fixed weights $w$, and combined with a fixed initial state $s_0$, equations (1) define a state machine[3]. This state machine can be seen as an adaptive policy or an online learning algorithm.

**Bayes-optimal policies as state machines**  Bayes-optimal policies have a natural interpretation as state machines following (1). Every such policy can be seen as a state-transition function $g$, which maintains sufficient statistics (i.e., a summary of the past experience that is statistically sufficient to implement the prediction/action strategy) and an output function $f$, which uses this information to produce optimal outputs (i.e., the best action or prediction given the observed trajectory) [11, 10]. For instance, to implement an optimal policy for a multi-armed bandit with (independent) Bernoulli rewards, it is sufficient to remember the number of successes and failures for each arm.

**Comparisons of state machines via simulation**  To compare the policies of a meta-trained and a Bayes-optimal agent in terms of their *computational structure*, we adapt a well-established methodology from the *state-transition systems* literature [28–30]. Specifically, we use the concept of *simulation* to compare state machines.

Formally, we have the following. A *trace* in a state machine is a sequence $s_0 x_1 s_1 \cdots x_T s_T$ of transitions. Since the state machines we consider are deterministic, a given sequence of inputs $x_1, \ldots, x_T$ induces a unique trace in the state machine. A deterministic state machine $M$ *simulates* another machine $N$, written $N \preceq M$, if every trace in $N$ has a corresponding trace in $M$ on which their output functions agree. More precisely, $N \preceq M$ if there exists a function $\phi$ mapping the states of $N$ into the states of $M$ such that the following two conditions hold:

- (*transitions*) for any trace $s_0 x_1 s_1 \cdots s_T$ in $N$, the transformed trace $\phi(s_0) x_1 \phi(s_1) \cdots \phi(s_T)$ is also a trace in $M$;
- (*outputs*) for any state $s$ of $N$ and any input $x$, the output of machine $N$ at $(x, s)$ coincides with the output of machine $M$ at $(x, \phi(s))$.

Intuitively, this means there is a consistent way of interpreting every state in $N$ as a state in $M$, such that every computation in $N$ can be seen as a computation in $M$. When both $M \preceq N$ and $N \preceq M$ hold, then we consider both machines to be computationally equivalent.

## 3   Methods

### 3.1   Tasks and agents

**Tasks**   Since our aim is to compare against Bayes-optimal policies, we consider 10 prediction and 4 reinforcement learning tasks for which the Bayes-optimal solution is analytically tractable. All tasks are episodic ($T = 20$ time steps), and the task parameters $\theta$ are drawn from a prior distribution $p(\theta)$ at the beginning of each episode. A full list of tasks is shown in Figure 4 and details are discussed in Appendix A.

In prediction tasks the goal is to make probabilistic predictions of the next observation given past observations. All observations are drawn i.i.d. from an observational distribution. To simplify the computation of the optimal predictors, we chose observational distributions within the exponential family that have simple conjugate priors and posterior predictive distributions, namely: Bernoulli, categorical, exponential, and Gaussian. In particular, their Bayesian predictors have finite-dimensional sufficient statistics with simple update rules [31–33].

In reinforcement learning tasks the goal is to maximise the discounted cumulative sum of rewards in two-armed bandit problems [34]. We chose bandits with rewards that are Bernoulli- or Gaussian-distributed. The Bayes-optimal policies for these bandit tasks can be computed in polynomial time by pre-computing *Gittins indices* [35, 34, 36]. Note that the bandit tasks, while conceptually simple, already require solving the exploration versus exploitation problem [37].

**RNN meta-learners**   Our RNN meta-learners consist of a three-layer network architecture: one fully connected layer (the encoder), followed by one LSTM layer (the memory), and one fully connected layer (the decoder) with a linear readout producing the final output, namely the parameters of the predictive distribution for the prediction tasks, and the logits of the softmax action-probabilities for the bandit tasks respectively. The width of each layer is the same and denoted by $N$. We selected[4] $N = 32$ for prediction tasks and $N = 256$ for bandit tasks. Networks were trained with BPTT [23, 24] and Adam [38]. In prediction tasks the loss function is the log-loss of the prediction. In bandit tasks the agents were trained to maximise the return (i.e., the discounted cumulative reward) using the Impala [39] policy gradient algorithm. See Appendix B.2 for details on network architectures and training.

### 3.2   Behavioral analysis

The aim of our behavioural analysis is to compare the input-output behaviour of a meta-learned (RNN) and a Bayes-optimal agent (Opt). For prediction tasks, we feed the same observations to both agent

types and then compute their dissimilarity as the sum of the KL-divergences of the instantaneous predictions averaged over $K$ trajectories, that is,

$$d(\text{Opt}, \text{RNN}) = \frac{1}{K} \sum_{k=1}^{K} \sum_{t=1}^{T} D_{\text{KL}}\big(\pi_t^{\text{Opt}} \big\| \pi_t^{\text{RNN}}\big). \tag{2}$$

Bandit tasks require a different dissimilarity measure: since there are multiple optimal policies, we cannot compare action probabilities directly. A dissimilarity measure that is invariant under optimal policies is the empirical reward difference:

$$d(\text{Opt}, \text{RNN}) = \Big| \frac{1}{K} \sum_{k=1}^{K} \sum_{t=1}^{T} (r_t^{\text{Opt}} - r_t^{\text{RNN}}) \Big| \tag{3}$$

where $r^{\text{Opt}}$ and $r^{\text{RNN}}$ are the empirical rewards collected during one episode. This dissimilarity measure only penalises policy deviations that entail reward differences.

### 3.3  Convergence analysis

In our convergence analysis we investigate how the behaviour of meta-learners changes over the course of training. To characterise how a single RNN training run evolves, we evaluate the behavioural dissimilarity measures (Section 3.2), which compare RNN behaviour against Bayes-optimal behaviour, across many checkpoints of a training run. Additionally we study the RNN behaviour across *multiple* training runs, which allows us to characterise convergence towards the Bayes-optimal solution. For this we use several RNN training runs (same architecture, different random initialisation), and at fixed intervals during training we compute pairwise behavioural distances between all meta-learners and the Bayes-optimal agent. The behavioural distance is computed using the Jensen-Shannon divergence[5] for prediction tasks and the absolute value of the cumulative regret for bandits. We visualise the resulting distance matrix in a 2D plot using multidimensional scaling (MDS) [40].

### 3.4  Structural analysis

We base our structural analysis on the idea of simulation introduced in Section 2. Since here we also deal with continuous state, input, and output spaces, we relax the notion of simulation to *approximate simulation*:

- (*Reference inputs*) As we cannot enumerate all the traces, we first sample a collection of input sequences from a reference distribution and then use the induced traces to compare state machines.

- (*State and output comparison*) To assess the quality of a simulation, we first learn a map $\phi$ that embeds the states of one state machine into another, and then measure the dissimilarity. To do so, we introduce two measures of dissimilarity $D_s$ and $D_o$ to evaluate the state and output dissimilarity respectively. More precisely, consider assessing the quality of a state machine $M$ simulating a machine $N$ along a trace induced by the input sequence $x_1 \cdots x_T$. Then, the quality of the state embedding $D_s$ is measured as the mean-squared-error (MSE) between the embedded states $\phi(\mathcal{S}_N) \subset \mathcal{S}_M$ and the states $\mathcal{S}_M$ of $M$ along the trace. Similarly, the quality of the output simulation $D_o$ is measured as the dissimilarity between the outputs generated from the states $\mathcal{S}_N$ and $\phi(\mathcal{S}_N)$ along the trace, that is, before and after the embedding respectively.

In practice, we evaluate how well e.g. a meta-learned agent simulates a Bayes-optimal one by first finding an embedding $\phi$ mapping Bayes-optimal states into meta-learned states that minimises the state dissimilarity $D_s$, and then using said embedding to compute the output dissimilarity $D_o$. The mapping $\phi$ is implemented as an MLP—see details in Appendix C. We use (2) and (3) as output dissimilarity measures $D_o$ in prediction and bandit tasks respectively.

Our approach is similar in spirit to [41], but adapted to work in continuous observation spaces.

# 4 Results

## 4.1 Behavioral Comparison

To compare the behavior between meta-learned and Bayes-optimal agents, we contrast their outputs for the same inputs. Consider for instance the two agents shown in Figure 1. Here we observe that the meta-learned and the Bayes-optimal agents behave in an almost identical manner: in the prediction case (Figure 1a), the predictions are virtually indistinguishable and approach the true probabilities; and in the bandit case (Figure 1b) the cumulative regrets are essentially the same[6] whilst the policy converges toward pulling the best arm.

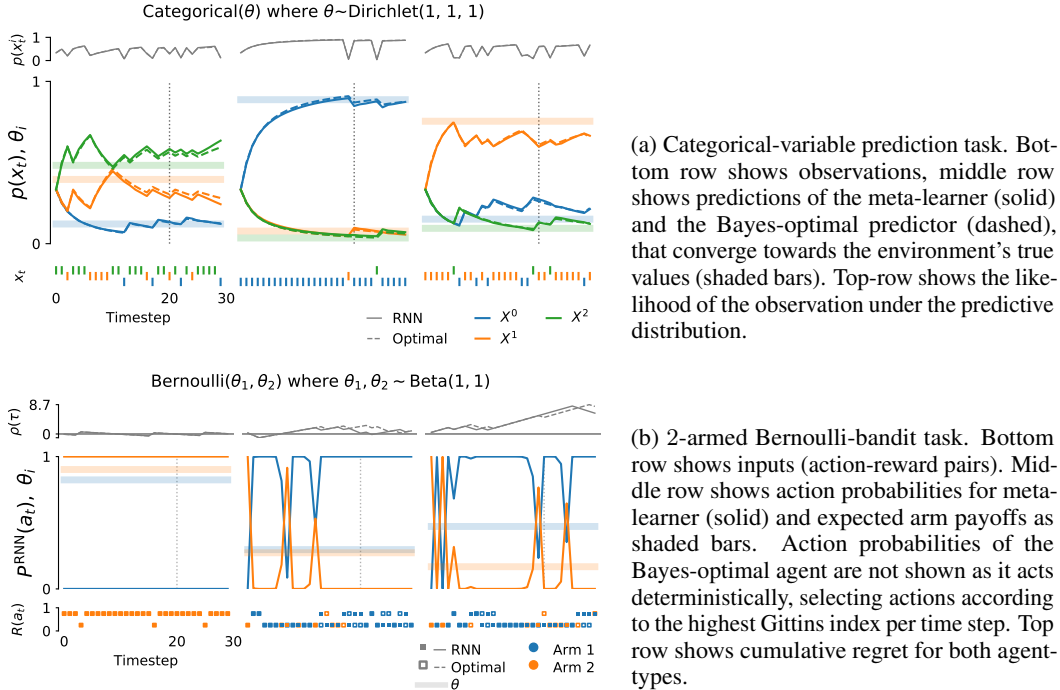

(a) Categorical-variable prediction task. Bottom row shows observations, middle row shows predictions of the meta-learner (solid) and the Bayes-optimal predictor (dashed), that converge towards the environment's true values (shaded bars). Top-row shows the likelihood of the observation under the predictive distribution.

(b) 2-armed Bernoulli-bandit task. Bottom row shows inputs (action-reward pairs). Middle row shows action probabilities for meta-learner (solid) and expected arm payoffs as shaded bars. Action probabilities of the Bayes-optimal agent are not shown as it acts deterministically, selecting actions according to the highest Gittins index per time step. Top row shows cumulative regret for both agent-types.

Figure 1: Illustrative behavioral comparison of a meta-learned agent and the Bayes-optimal agent on 3 episodes (same environment random seed for both agents). Meta-learned agents were trained with only 20 time-steps; thus these results illustrate that the RNN generalizes to 30 time-steps.

To quantitatively assess behavioral similarity between the meta-learners and the Bayes-optimal agents, we use the measures introduced in Section 3.2, namely (2) for prediction tasks and (3) for bandit tasks. For each task distribution, we averaged the performance of 10 meta-learned agents. The corresponding results in Figure 4a show that the trained meta-learners behave virtually indistinguishably from the Bayes-optimal agent. Results for reduced-memory agents, which cannot retain enough information to perform optimally, are shown in Appendix D.7.

## 4.2 Convergence

We investigate how the behavior of meta-learners changes over the course of training. Following the methodology introduced in Section 3.3 we show the evolution of behavior of a single training run (top-left panels in Figure 2a, 2b). These results allow us to evaluate the meta-learners' ability to pick up on the environment's prior statistics and perform Bayesian evidence integration accordingly. As training progresses, agents learn to integrate the evidence in a near-optimal manner over the entire course of the episode. However, during training the improvements are not uniform throughout the episode. This 'staggered' meta-learning, where different parts of the task are learned progressively, resembles results reported for meta-learners on nonlinear regression tasks in [42].

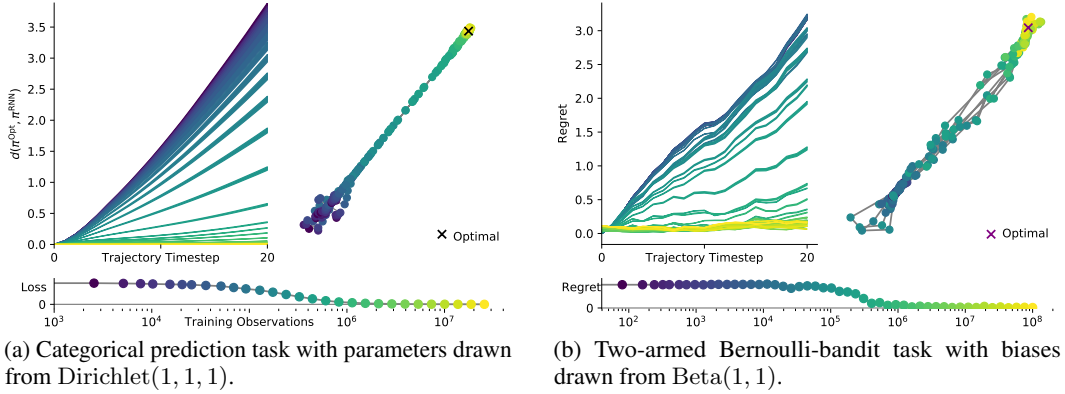

(a) Categorical prediction task with parameters drawn from $\mathrm{Dirichlet}(1, 1, 1)$.

(b) Two-armed Bernoulli-bandit task with biases drawn from $\mathrm{Beta}(1, 1)$.

Figure 2: Policies evolve similarly towards the Bayes-optimal policy over the course of training for both the prediction (a) and the bandit task (b). Panels in each subfigure show, (clockwise from top left): evolution of the within-episode dissimilarity from the Bayes-optimal policy, averaged over $500$ trajectories; the evolution of 10 policies for different training runs (multidimensional scaling visualisation of pairwise behavioural distances; each curve is a separate run); and the training curves for the log-loss and regret respectively.

We also compared behavior across multiple training runs (top-right panels in Figure 2a, 2b). Overall the results indicate that after some degree of heterogeneity early in training, all meta-learners converge in a very similar fashion to the Bayes-optimal behavior. This is an empirical confirmation of the theoretical prediction in [10] that the Bayes-optimal solution is the fixed-point of meta-learner training. Appendix D.5 shows the convergence results for all tasks.

## 4.3 Structural Comparison

In this section we analyze the computational structure of the meta-learner who uses its internal state to store information extracted from observations required to act.

Following the discussion in Section 3.4, we determine the computational similarity of the meta-learning and Bayes-optimal agents via simulation. Our analysis is performed by projecting and then whitening both the RNN state (formed by concatenating both the cell- and hidden-states of the LSTM) and the Bayes-optimal state onto the first $n$ principal components, where $n$ is the dimensionality of the Bayes-optimal state/sufficient statistics. We find that these few components suffice to explain a large fraction of the variance of the RNN agent's state—see Appendix D.2. We then regress an MLP-mapping $\phi$ from one (projected) agent state onto the other and compute $D_s$ and $D_o$. Importantly, this comparison is only meaningful if we ensure that both agents were exposed to precisely the same input history. This is easily achieved in prediction tasks by fixing the environment random seed. In bandit tasks we ensure that both agents experience the same action-reward pairs by using the trained meta-learner to generate input streams that are then also fed into the Bayes-optimal agent.

Figure 3 illustrates our method for assessing the computational similarity. We embedded[7] the state space of the Bayes-optimal agent into the state space of the meta-learned agent, and then we calculated the output from the embedded states. This embedding was also performed in the reverse direction. Visual inspection of this figure suggests that the meta-learned and the Bayes-optimal agents perform similar computations, as the panels resemble each other both in terms of states and outputs. We observed similar results for all other tasks (Appendix D.5). In contrast, we have observed that the computational structure of *untrained* meta-learners does not resemble the one of Bayes-optimal agents (Appendix D.1).

The quantitative results for the structural comparison for all tasks across 10 repetitions of meta-training are shown in Figure 4. We find that for the trained meta-learner state-dissimilarity $D_s$ is low in almost all cases. In bandit tasks, $D_s$ tends to be slightly larger in magnitude which is somewhat expected since the RNN-state dimensionality is much larger in bandit tasks. Additionally there is often no significant difference in $D_s$ between the untrained and the final agent—we suspect this to be an artefact of a reservoir effect [43] (see Discussion). The output-dissimilarity $D_o$ is low for both

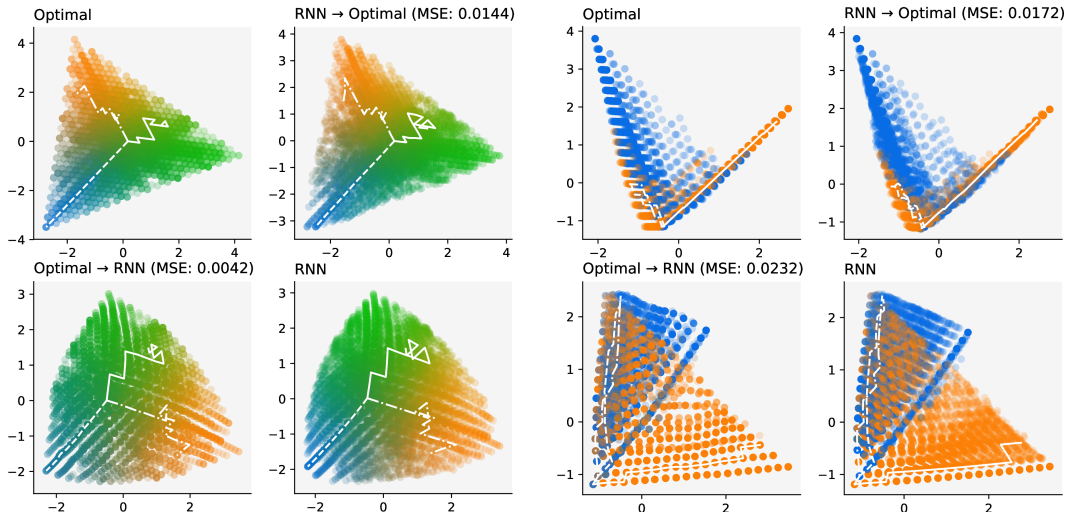

(a) Categorical prediction task with parameters drawn from $\mathrm{Dirichlet}(1, 1, 1)$. The colors indicate the prediction probabilities emitted in each state. Three episode rollouts are shown.

(b) Two-armed Bernoulli-bandit task with biases drawn from $\mathrm{Beta}(1, 1)$. The colors indicate the action-probabilities emitted in each state. Three episode rollouts are shown.

Figure 3: Structural comparison. Each sub-figure depicts: two agent state spaces, namely of the Bayes-optimal (top-left) and RNN states (bottom-right), projected onto the first two principal components; and two simulations, i.e., the learned embeddings from the RNN into the Bayes-optimal states (top-right) and from the Bayes-optimal into the RNN states (bottom-left). The scores in the simulations indicate the MSE of the learned regression. The outputs emitted in each state are color-coded. Note that the color-codings in the simulations result from evaluating the output at the (potentially high-dimensional) embedded state (see Section 3.4). White lines indicate the same three episodes as shown in Figure 1.

task types for $\mathrm{RNN} \to \mathrm{Opt}$, but not in the reverse direction. This indicates that the meta-learners are very well simulated by the Bayes-optimal agents, since both the state dissimilarity $D_s$ and the output dissimilarity $D_o$ are almost negligible. In the reverse direction however, we observe that the meta-learned solutions do not always simulate the Bayes-optimal with high accuracy, as seen by the non-negigible output dissimilarity $D_o$. We believe that this is because the sufficient statistics learned by the meta-learners are not minimal.

## 5    Discussion and conclusions

In this study we investigated whether memory-based meta-learning leads to solutions that are behaviourally and structurally equivalent to Bayes-optimal predictors. We found that behaviorally the Bayes-optimal solution constitutes a fixed-point of meta-learner training dynamics. Accordingly, trained meta-learners behave virtually indistinguishable from Bayes-optimal agents. We also found structural equivalence between the two agent types to hold to a large extent: meta-learners are well simulated by Bayes-optimal agents, but not necessarily vice versa. This failure of simulation is most likely a failure of injectivity: if a single state in one agent must be mapped to two distinct states in another then simulation is impossible. This occurs when two trajectories lead to the same state in one agent but not another (for instance if exchangeability has not been fully learned). We suspect that RNN meta-learners represent non-minimal sufficient statistics as a result of training. For instance, for Bernoulli prediction tasks the input sequences *heads-tails-heads*, and *tails-heads-heads* induce the same minimal sufficient statistics and thus lead to precisely the same internal state in the Bayes-optimal agent, but might lead to different states in the RNN agent. From a theoretical point of view this is not unexpected, since there is no explicit incentive during RNN training that would force representations to be minimal. Note that overly strong regularization can reduce the RNN's effective capacity to a point where it can no longer represent the number of states required by Bayes-optimal solution, which of course strictly rules out computational equivalence.

A related issue can be observed in bandit tasks: even untrained meta-learners show low state-dissimilarity. We hypothesize that this is due to a "reservoir effect" [43], that is the dynamics of

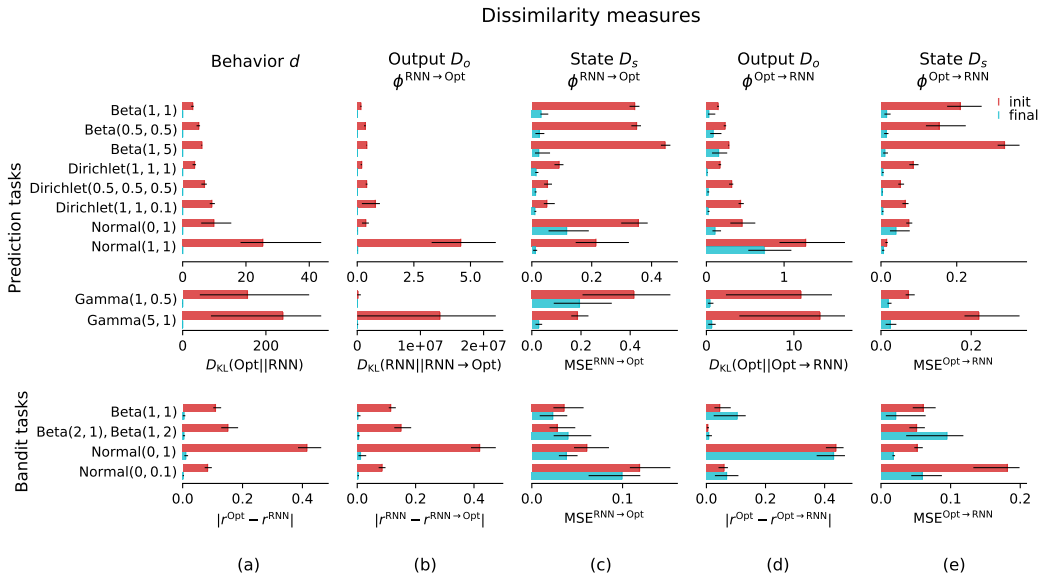

Figure 4: Behavioral and structural comparison for all tasks. Figure shows similarity measures across $K = 500$ episodes of length $T = 20$, and 10 different training runs of the meta-learner (bars show median over training-runs, error bars denote 5-95 quantiles). 'init' denotes the untrained meta-learner, 'final' denotes evaluation at the end of training. Numerical results are shown in Figure 7 in the Appendix. Column a: Behavioral dissimilarity between meta-learned agent and Bayes-optimal agent (see Section 4.1). Columns b & c: State- and Output-dissimilarity for $\mathrm{RNN} \rightarrow \mathrm{Opt}$. Columns d & e: State- and Output-dissimilarity for $\mathrm{Opt} \rightarrow \mathrm{RNN}$. Low values of the state- and output-dissimilarity measures (simultaneously) indicate that the state machines implemented by $\mathrm{RNN}$ and $\mathrm{Opt}$ are structurally equivalent.

high-dimensional untrained RNNs are highly likely to map each input history to a unique trajectory in memory space. Accordingly, the untrained RNN "memorizes" inputs perfectly—a verbose representation of the task's sufficient statistics.

Our work contributes to understanding the structure and computations implemented by recurrent neural networks. Focusing analysis on computational equivalence, as in our work, opens up the future possibility of separating different, heterogeneous agents into meaningful sets of equivalent classes, and study universal aspects of these agent-classes.

## 5.1   Related work

In this paper we study memory-based meta-learning through a Bayesian lens, showing that meta-learning objectives naturally induce Bayes-optimal behaviour at convergence. A number of previous works have attempted to devise new recurrent architectures to perform Bayes filtering in a number of settings, including time series prediction [44], state space modelling [45], and Kalman filtering [46]. Other previous work has attempted to improve memory-based meta-learners' abilities by augmenting them with a memory, and using weights which adapt at different speeds [47, 48].

Another approach to meta-learning is optimiser-based meta-learning such as MAML [49]. In optimiser-based meta-learning models are trained to be able to adapt rapidly to new tasks via gradient descent. MAML has been studied from a Bayesian perspective, and shown to be a hierarchical Bayesian model [50]. Recent work suggests that solutions obtained by optimiser-based meta-learning might be more similar to those from memory-based meta-learning than previously thought [51] .

In this paper we relate memory-based meta-learning to finite-state automata which track sufficient statistics of their inputs. The field of computational mechanics [52] studies predictive automata (known as $\varepsilon$-*machines*) which track the state of a stochastic process in order to predict its future states. The states of $\varepsilon$-machines are referred to as causal states, and have recently been used to augment recurrent agents in POMDPs [53]. Finite-state automata have also been considered as a model for decision-making agents in the *situated automata* work of Rosenschein and Kaelbling [54, 55]. The states of situated automata track logical propositions about the state of the world instead of having a probabilistic interpretation, but are naturally suited to goal-directed agents.

There is considerable work on understanding recurrent neural networks on natural language tasks [56], and in neuroscience [57–59], e.g. how relations between multiple trained models can illuminate computational mechanisms [15], and the dynamics involved in contextual processing [13]. Computational analysis of internal dynamics of reinforcement learning agents has received less attention in the literature, though there are some notable examples: a multi-agent setting [18] and Atari games [60]. Using a related formalism to our approach, the authors of [61] extract minimal finite-state machines (FSM) from the internal dynamics of Atari-agents. However their focus is on extracting small human-interpretable FSM, whereas we compare the computational structure of two agents in a fully automated, quantitative fashion.

In recent years a diverse range of tools to allow interpretability and explainability of deep networks have been developed, including saliency maps [62–67], manual dissection of individual units [17, 16, 68] and training explainable surrogate models to mimic the output of deep networks [69, 61]. Although our focus here is different - we seek to establish how a broad class of architectures behaves on a family of tasks, rather than explaining a specific network - the closest parallel is with the use of surrogate explainable models. In this case, the Bayes-optimal agent serves as an understood model, and we relate its (well-understood) behaviour to that of the meta-trained agent.

**Scope and limitations**    We performed our empirical comparison on a range of tasks where optimal solutions are analytically and computationally tractable. The latter is typically no longer true in more complex tasks and domains. However, the simulation methodology used in this paper could be useful to compare agent-types against each other in more general settings, as it does not rely on either agent being Bayes-optimal. While most aspects of our methodology scale up well to more complex agents, the main difficulty is generating reference trajectories that cover a large (enough) fraction of possible experiences. Finally, our results show that when optimal policies are in the search space, and training converges to those policies, then the resulting policy will be Bayes-optimal. In more complex cases, one or both of these assumptions may no longer hold. Further study is needed to understand the kind of suboptimal solutions that are generated by meta-learning in this case.

**Conclusions**    Our main contribution is to advance the understanding of RNN-based meta-learned solutions. We empirically confirm a recently published theoretical claim [10] that fully-converged meta-learners and Bayes-optimal agents are computationally equivalent. In particular, we showed that RNN meta-learners converge during training to the Bayes-optimal solution, such that trained meta-learners behave virtually indistinguishably from Bayes-optimal agents. Using a methodology related to the concept of *simulation* in theoretical computer science, we additionally show (approximate) structural equivalence of the state-machines implemented by the RNN meta-learners and the Bayes-optimal agent. Our results suggest that memory-based meta-learning will drive learned policies towards Bayes-optimal behaviour, and will converge to this behaviour where possible.

# 6    Broader Impact

Our work helps advance and verify the current understanding of the nature of solutions that meta-learning brings about (our empirical work focused on modern recurrent neural network architectures and training algorithms, but we expect the findings to qualitatively hold for a large range of AI systems that are trained through meta-learning). Understanding how advanced AI and ML systems work is of paramount importance for safe deployment and reliable operation of such systems. This has also been recognized by the wider machine-learning community with a rapidly growing body of literature in this emerging field of "Analysis and Understanding" of deep learning. While increased understanding is likely to ultimately also contribute towards building more capable AI systems, thus potentially amplifying their negative aspects, we strongly believe that the merits of understanding how these systems work clearly outweigh the potential risks in this case.

We argue that understanding meta-learning on a fundamental level is important, since meta-learning subsumes many specific learning tasks and is thought to play an important role for AI systems that generalize well to novel situations. Accordingly we expect meta-learning to be highly relevant over the next decade(s) in AI research and in the development of powerful AI algorithms and applications. In this work we also show a proof-of-concept implementation for analysis methods that might potentially allow one to separate (heterogeneous) agents into certain equivalence classes, which would allow to safely generalize findings about an individual agent to the whole equivalence class. We believe that this might open up interesting future opportunities to boost the generality of analysis methods and automatic diagnostic tools for monitoring of AI systems.

## Acknowledgments and Disclosure of Funding

We thank Jane Wang and Matt Botvinick for providing helpful comments on this work.

Funding in direct support of this work: no specific or third-party funding was received. All authors are full-time employed with DeepMind.

## Footnotes

[3]More precisely, a Mealy machine [25–27].

[4]Note that the (effective) network capacity needs to be large enough to at least represent the different states required by the Bayes-optimal solution. However, it is currently unknown how to precisely measure effective network capacity. We thus selected our architectures based on preliminary ablations that investigate convergence speed of training. See Appendix D.3 for details.

[5]The Jensen-Shannon divergence is defined as $D_{\text{JS}}(X\|Y) = \frac{1}{2}(D_{\text{KL}}(X\|M) + D_{\text{KL}}(Y\|M))$, where $M$ is the mixture distribution $(X + Y)/2$.

[6]Recall that the regret is invariant under optimal policies.

[7]The embeddings were implemented as MLPs having three hidden layers with either 64 (prediction) or 256 (bandits) neurons each.

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
