[Supplementary Material]

## Supplementary Material

## A    Task Details

There is a total of $14$ tasks, out of which $10$ are prediction and $4$ are bandit tasks.

**Prediction:**    The prediction tasks can be grouped according to their observational distributions:

- *Bernoulli:* The agent observes samples $x_t$ drawn from a Bernoulli distribution $\mathrm{Ber}(\theta)$. The prior distribution over the bias $\theta$ is given by a Beta distribution $\mathrm{Beta}(\alpha, \beta)$, where $\alpha > 0$ and $\beta > 0$ are the hyperparameters. We have three tasks with three respective prior distributions: $\mathrm{Beta}(1, 1)$, $\mathrm{Beta}(0.5, 0.5)$, and $\mathrm{Beta}(1, 5)$.

- *Categorical:* The agent observes samples $x_t$ drawn from a categorical distribution $\mathrm{Cat}(\vec{\theta})$ where $\vec{\theta} = [\theta_1, \theta_2, \theta_3]^T$. The prior distribution over the bias parameters $\vec{\theta}$ is given by a Dirichlet distribution $\mathrm{Dirichlet}(\vec{\alpha})$, where $\vec{\alpha} = [\alpha_1, \alpha_2, \alpha_3]^T$ are the concentration parameters. We have three categorical tasks with three respective prior distributions: $\mathrm{Dirichlet}(1, 1, 1)$, $\mathrm{Dirichlet}(1, 1, 0.1)$, and $\mathrm{Dirichlet}(0.5, 0.5, 0.5)$.

- *Exponential:* The agent observes samples $x_t$ drawn from an exponential distribution $\mathrm{Exp}(\lambda)$ where $\lambda > 0$ is the rate parameter. The prior distribution over the rate parameter $\lambda$ is given by a Gamma distribution $\mathrm{Gamma}(\alpha, \beta)$, where $\alpha > 0$ is the shape and $\beta > 0$ is the rate. We use two exponential prediction tasks: their priors are $\mathrm{Gamma}(1, 0.5)$ and $\mathrm{Gamma}(5, 1)$.

- *Gaussian:* The agent observes samples $x_t$ drawn from a Gaussian distribution $\mathrm{Normal}(\mu, 1/\tau)$, where $\mu$ is an unknown mean and $\tau$ is a known precision. The prior distribution over $\mu$ is given by a Gaussian distribution $\mathrm{Normal}(m, 1/p)$, where $m$ and $p$ are the prior mean and precision parameters. We have two Gaussian prediction tasks: their priors are $\mathrm{Normal}(0, 1)$ and $\mathrm{Normal}(1, 1)$ and their precisions $\tau = 1$ and $\tau = 5$ respectively.

A prediction task proceeds as follows. As a concrete example, consider the Bernoulli prediction case—other distributions proceed analogously. In the very beginning of each episode, the bias parameter $\theta$ is drawn from a fixed prior distribution $p(\theta) = \mathrm{Beta}(1, 1)$. This parameter is never shown to the agent. Then, in each turn $t = 1, 2, \ldots, T = 20$, the agent makes a probabilistic prediction $\pi_t$ and then receives an observation $x_t \sim p(x|\theta) = \mathrm{Ber}(\theta)$ drawn from the observational distribution. This leads to a prediction loss given by $-\log(\pi_t(x_t))$, where $\pi_t(x_t)$ is the predicted probability of the observation $x_t$ at time $t$. Then the next round starts.

**Bandits:**    As in the prediction case, the two-armed bandit tasks can also be grouped according to their reward distributions:

- *Bernoulli:* Upon pulling a lever $a \in \{1, 2\}$, the agent observes a reward sampled from a Bernoulli distribution $\mathrm{Ber}(\theta_a)$, where $\theta_a$ is the bias of arm $a$. The prior distribution over each arm bias is given by a Beta distribution as in the prediction case. We have two Bernoulli bandit tasks: the first draws both biases from $\mathrm{Beta}(1, 1)$, and the second from $\mathrm{Beta}(2, 1)$ and $\mathrm{Beta}(1, 2)$ respectively.

- *Gaussian:* Upon pulling a lever $a \in \{1, 2\}$, the agent observes a reward sampled from a Gaussian distribution $\mathrm{Normal}(\mu, \tau)$, where $\mu$ and $\tau$ are the unknown mean and the known precision of arm $a$ respectively. As in the prediction case, the prior distribution over each arm mean is given by a Normal distribution. We have two Gaussian bandit tasks: the first with precision $\tau = 1$ and prior $\mathrm{Normal}(0, 1)$ for both arms; and the second with precision $\tau = 1$ and prior $\mathrm{Normal}(0, 0.1)$.

The interaction protocol for bandit tasks is as follows. For concreteness we pick the first Bernoulli bandit—but other bandits proceed analogously. In the very beginning of each episode, the arm biases $\theta_1$ and $\theta_2$ are drawn from a fixed prior distribution $p(\theta) = \mathrm{Beta}(1, 1)$. These parameters are never shown to the agent. Then, in each turn $t = 1, 2, \ldots, T$, the agent pulls a lever $a \sim \pi_t$ from its policy at time $t$ and receives a reward $r_t \sim p(r|\theta_a) = \mathrm{Ber}(\theta_a)$ drawn from the reward distribution. Then the next round starts. The agent's return is the discounted sum of rewards $\sum_t \gamma^t r_t$ with discount factor $\gamma = 0.95$.

Table 1: Prediction rules for Bayes-optimal agents

| Observation | Prior | Update | Posterior Predictive |
|---|---|---|---|
| Bernoulli($\theta$) | Beta($\alpha, \beta$) | $\alpha \leftarrow \alpha + x; \beta \leftarrow \beta + (1 - x)$ | Bernoulli($\frac{\alpha}{\alpha + \beta}$) |
| Categorical($\vec{\theta}$) | Dirichlet($\alpha_1, \alpha_2, \alpha_3$) | $\alpha_x \leftarrow \alpha_x + 1$ | Categorical($\frac{\alpha_i}{\sum_j \alpha_j}$) |
| Normal($\mu, 1/\tau$) | Normal($m, 1/p$) | $m \leftarrow \frac{pm + \tau x}{p + \tau}; p \leftarrow p + \tau$ | Normal($m, \frac{1}{p} + \frac{1}{\tau}$) |
| Exponential($\lambda$) | Gamma($\alpha, \beta$) | $\alpha \leftarrow \alpha + 1, \beta \leftarrow \beta + x$ | Lomax($\alpha, \beta$) |

## B  Agent Details

### B.1  Bayes-optimal agents

Our Bayes-optimal agents act and predict according to the standard models in the literature. We briefly summarize this below.

**Prediction:**  A Bayes-optimal agent makes predictions by combining a prior with observed data to form a posterior belief. Consider a Bernoulli environment that generates observations according to Bernoulli($\theta$), where in each episode $\theta \sim \text{Beta}(1,1)$. In each turn $t$, the agent makes a prediction according to the *posterior predictive distribution*

$$p(x_t | x_{<t}) = \int p(x_t | \theta) p(\theta | x_{<t}) d\theta, \tag{4}$$

where the prior $p(\theta | x_{<t})$ is the posterior of the previous turn (in the first step the agent uses its prior, which, for the optimal agent, coincides with the environment's prior). Subsequently, the agent receives an observation $x_t$, which and updates its posterior belief:

$$p(\theta | x_{\leq t}) \propto p(\theta | x_{<t}) p(x_t | \theta). \tag{5}$$

Note that for the distributions used in our prediction tasks, the posterior can be parameterized by a small set of values: the minimal sufficient statistics (which compress the whole observation history $x_{<t}$ into the minimal amount of information required to perform optimally).

For a Bernoulli predictor, the posterior predictive (4) is equal to

$$p(x_t | x_{<t}) = p(x_t | \alpha, \beta) = \text{Ber}(\tfrac{\alpha}{\alpha + \beta}).$$

where, $\alpha$ and $\beta$ are the sufficient statistics. The posterior belief is given by

$$p(\theta | x_{\leq t}) = p(\theta | \alpha', \beta') = \text{Beta}(\alpha', \beta'),$$

where $\alpha' = \alpha + x_t$ and $\beta' = \beta + (1 - x_t)$ are the hyperparameters updated by the observation $x_t$. For a full list of update and prediction rules, see Table 1.

**Bandits:**  A Bayes-optimal bandit player maintains beliefs for each arm's distribution over the rewards. For instance, if the rewards are distributed according to a Bernoulli law, then the agent keeps track of one $(\alpha, \beta)$ sufficient-statistic pair per arm. The optimal arm to pull next is then given by

$$a^* = \arg\max_a Q(a | \alpha_1, \beta_1, \alpha_1, \beta_1), \tag{6}$$

where the Q-value is recursively defined as

$$Q(a | \alpha_1, \beta_1, \alpha_1, \beta_1) := 0 \qquad\qquad \text{if } t = T$$
$$Q(a | \alpha_1, \beta_1, \alpha_1, \beta_1) := \sum_r p(r | \alpha_a, \beta_a) \Big\{ r + \max_{a'} \gamma Q(a' | \alpha_1', \beta_1', \alpha_2', \beta_2') \Big\} \quad \text{if } t < T \tag{7}$$

and where $\alpha_1', \beta_1', \alpha_2', \beta_2'$ are the hyperparameters for the next step, updated in accordance to the action taken and the reward observed. Computing (6) naively is computationally intractable. Instead, one can pre-compute *Gittins indices* in polynomial time, and use them as a replacement for the Q-values in (6) [35, 34, 36]. In particular, we have used the methods presented in [36] to compute Gittins indices for the Bernoulli- and Gaussian-distributed rewards.

## B.2  RNN agents

**Prediction:**  We trained agents on the prediction tasks (episode length $T = 20$ steps) using supervised learning with a batch size of 128 using BPTT unroll of 20 timesteps, and a total training duration of $1e7$ steps. We used the Adam optimizer with learning rate $10^{-4}$, parameters $\beta_1 = 0.9$, $\beta_2 = 0.999$, and gradients clipped at magnitude 1. Networks were initialised with weights drawn from a truncated normal with standard deviation $1/\sqrt{N_{\text{in}}}$, where $N_{\text{in}}$ is the size of the input layer. We use the following output-parametrization: Bernoulli-predictions - single output corresponding to the log-probability (of observing "heads"); Categorical predictions - 3-D outputs corresponding to prediction logits; Normal predictions - 2 linear outputs, one for mean and one for log-precision; Exponential predictions - 2 linear outputs, one for $\log \alpha$ and one for $\log \beta$.

**Bandit:**  We trained the reinforcement learners on bandit tasks (episode length $T = 20$ steps) with the Impala algorithm [39] using a batch size of 16 and discount factor $\gamma = 0.95$ for a total number of $1e8$ training steps. The BPTT unroll length was 5 timesteps, and the learning rate was $2.5 \times 10^{-5}$. We used an entropy penalty of $0.003$ and value baseline loss scaling of $0.48$; i.e.,, the training objective was $\mathcal{L}_{\text{VTrace}} + 0.003\mathcal{L}_{\text{Entropy}} + 0.48\mathcal{L}_{\text{Value}}$. We used the same initialisation scheme as for the prediction tasks. RNN outputs in all bandit tasks were 2-dimensional action logits (one for each arm). Bandit agents are trained to minimize empirical ("sampled") cumulative discounted rewards. For our behavioral and output dissimilarity measures we report expected reward instead of sampled reward (using the environment's ground-truth parameters to which the agent does not have access to)—this reduces the impact of sampling noise on our estimates.

## C  Structural Comparison Details

We implement the map $\phi$ from RNN agent states $\mathcal{S}_N$ to optimal agent states $\mathcal{S}_M$ using an MLP with three hidden layers, each of size 64 (prediction tasks where the RNN state is 64-dimensional) or 256 (bandit tasks where the RNN state is 512-dimensional), with ReLu activations. We first project the high-dimensional RNN agent state space down to a lower-dimensional representation using PCA. The number of principal components is set to match the dimension of the minimal sufficient statistics required by the task. We trained the MLP using the Adam optimiser with learning rate $0.001$, $\beta_1 = 0.9$, $\beta_2 = 0.999$ and batch size 200. The training set consisted of data from 500 roll-outs—all results we report were evaluated on 500 held out test-trajectories.

State dissimilarity $D_s$ is measured by providing the same inputs to both agents (same observations in prediction tasks, and action-reward pairs from a reference trajectory[8] in bandit tasks), and then taking the mean-squared error between the (PCA-projected) original states and the mapped states (compare Figure 3 in the main paper). Output dissimilarity is computed by comparing the output produced by the original agent with the output produced after projecting the original agent state into the "surrogate" agent and evaluating the output. Note that the last step requires inverting the PCA projection in order to create a "valid" state in the surrogate agent. For the optimal agent the PCA is invertible since its dimensionality is the same as the agent's state (i.e., the PCA on the optimal agent simply performs a rotation and whitening). On the RNN agent, we use the following scheme: we construct an invertible PCA projection as well, which requires having the same number of components as the internal state's dimensionality. Then, to implant a state from the Bayes-optimal agent the first $n$ components are set according to the mapping $\phi$, all other principal components are set to their mean-value (across 500 episodes).

## D  Additional Results

### D.1  PCA for untrained meta-learner

Figure 5 shows the principal component projection and approximate simulation (mapping the state of one agent onto the other and computing the resulting output) for meta-learner after random initialization, without any training. Results for the trained agent (at the end of the training run) are shown in Figure 3 in the main paper.

(a) Categorical-variable prediction task Dirichlet$(1, 1, 1)$. Colors indicate the output-probabilities (=posterior predictive dist.) for the corresponding state. Lines correspond to the three episodes shown in Figure 1. Dimensionality of $s_t^{\mathrm{rnn}}$ is 64. MLP-regressor $\phi$ has three hidden layers with 64 neurons each.

(b) 2-armed Bernoulli-bandit task $\sim \mathrm{Beta}(1, 1)$. Colors indicate the output-probabilities (=action probabilities) for the corresponding state. Lines correspond to the three episodes shown in Figure 1. Dimensionality of $s_t^{\mathrm{rnn}}$ is 512. MLP-regressor $\phi$ has three hidden layers with 256 neurons each.

Figure 5: Structural comparison for **untrained agent** (compare Figure 3 in main paper). Each sub-figure shows: (i - top left) Projection of Bayes-optimal state onto first two principal components, (iv - bottom right) projection of RNN state onto first two principal components, (ii - top right) learned regression from (iv) to (i), (iii - bottom left) learned regression from (i) to (iv). Scores in panels (ii) and (iii) indicate the mean-squared-error (MSE) of the learned regression (map $\phi$ was trained on training data, plots and numerical results show evaluation on held-out test-data—500 data-points for training and test respectively).

## D.2 Variance explained by PC projections

Table 2 shows the variance explained when projecting the RNN state onto the first $n$ principal components, which is the first step of our structural analysis ($n$ is the dimensionality of the tasks' minimal sufficient statistics, and is between 2 and 4 dimensions)—see Section 4.3. Numbers indicate the variance explained by projecting 500 trajectories of length $T = 20$ onto first $n$ principal components. Large number indicate that most of the variance in the data is captured by the PCA projection, which is the case for us in all tasks.

## D.3 Preliminary architecture sweeps

The meta-learners in our main experiments are three-layer RNNs (a fully connected encoder, followed by a LSTM layer and a fully connected decoder). Each layer has the same width $N$ which was selected by running preliminary architecture sweeps (on a subset of tasks), shown in Figure D.3. Generally we found that smaller RNNs suffice to successfully train on the prediction tasks compared to the RNN tasks. For instance a layer-width of 3 would suffice in principle to perform well on the prediction tasks (not that the maximum dimensionality of the minimal sufficient statistics is also exactly 3). However, we found that the smallest networks also tend to require more iterations to converge, with more noisy convergence in general. We thus selected $N = 32$ for prediction tasks (leading to a 64-dimensional RNN state, which is the concatenation of cell- and hidden-states) as a compromise between RNN-state dimensionality, runtime-complexity and iterations required for training to converge robustly (in our main experiments we train prediction agents for $1e7$ steps, and bandit agents for $1e8$ steps). Using similar trade-offs we chose $N = 256$ for bandit tasks (leading to a 512-dimensional RNN state).

Table 2: Variance of RNN-state explained by PCA projection.

|  | Task | at initialization | after training |
|---|---|---|---|
| Prediction tasks | $\text{Beta}(1,1)$ | 0.98 | 0.94 |
|  | $\text{Beta}(0.5, 0.5)$ | 0.98 | 0.92 |
|  | $\text{Beta}(1,5)$ | 0.98 | 0.96 |
|  | $\text{Dirichlet}(0.5, 0.5, 0.5)$ | 0.93 | 0.96 |
|  | $\text{Dirichlet}(1,1,1)$ | 0.93 | 0.95 |
|  | $\text{Dirichlet}(1,1,0.1)$ | 0.94 | 0.96 |
|  | $\text{Gamma}(1, 0.5)$ | 0.95 | 0.97 |
|  | $\text{Gamma}(5,1)$ | 0.97 | 0.96 |
|  | $\text{Normal}(0,1)$ | 0.95 | 0.88 |
|  | $\text{Normal}(1,1)$ | 0.97 | 0.94 |
| Bandits | $\text{Beta}(1,1)$ | 0.97 | 0.96 |
|  | $\text{Beta}(2,1), \ \text{Beta}(1,2)$ | 0.98 | 0.97 |
|  | $\text{Normal}(0,1)$ | 0.94 | 0.92 |
|  | $\text{Normal}(0, 0.1)$ | 0.95 | 0.90 |

(a) Subset of prediction tasks. Lines show difference between RNN and Bayes-optimal log-loss, averaged over 10 training runs.

(b) Bandit tasks. Lines show mean reward computed over the last 10k steps (rolling average) for a single training run.

Figure 6: Architecture sweeps.

## D.5 Structural comparison

We report the structural comparison plots for all the tasks. These were generated using the same methodology as in Figure 3. Figures 8, 9, and 10 show the comparisons for the prediction of discrete observations, prediction of continuous observations, and bandits respectively.

## D.6 Convergence analysis - additional results

Convergence plots for all our tasks (except the two exponential prediction tasks, where the KL-divergence estimation for the Lomax distribution can cause numerical issues that lead to bad visual results) are shown in Figure 11 and Figure 12. Note that our agents were trained with episodes of 20 steps, and the figures show how agents generalize when evaluated on episodes of 30 steps.

## D.4   Behavioral and structural comparison

Figure 7: Behavioral and structural comparison for all tasks—same as Figure 4 in main paper. Figure shows dissimilarity measures across 500 episodes of length $T = 20$, and 10 different training runs of the meta-learner. 'init' denotes the untrained meta-learner, 'final' denotes evaluation at the end of training. Colored bars show median across training runs (also given as numerical values on $y$-axis), error bars denote 5-95 quantiles (bold numbers indicate that upper end of 'final' error bar is strictly lower than lower end of 'init' error bar), vertical grey ticks indicate mean values (across training runs). **(a)** Behavioral dissimilarity between meta-learned agent and Bayes-optimal agent (see Section 4.1). **(b)**, **(c)** State- and Output-dissimilarity for RNN $\rightarrow$ Opt. **(d)**, **(e)** State- and Output-dissimilarity for Opt $\rightarrow$ RNN.

(a) Bernoulli($\theta$), $\theta \sim \text{Beta}(1, 1)$

(b) Bernoulli($\theta$), $\theta \sim \text{Beta}(0.5, 0.5)$

(c) Bernoulli($\theta$), $\theta \sim \text{Beta}(1, 5)$

(d) Categorical($\vec{\theta}$), $\vec{\theta} \sim \text{Dir}(1, 1, 1)$

(e) Categorical($\vec{\theta}$), $\vec{\theta} \sim \text{Dir}(1, 1, 0.1)$

(f) Categorical($\vec{\theta}$), $\vec{\theta} \sim \text{Dir}(0.5, 0.5, 0.5)$

Figure 8: Structural comparison I. Prediction probabilities are color-coded.

(a) Exponential($\lambda$), $\lambda \sim$ Gamma$(1, 0.5)$

(b) Exponential($\lambda$), $\lambda \sim$ Gamma$(5, 1)$

(c) Normal($\mu, 1$), $\mu \sim$ Normal$(0, 1)$

(d) Normal($\mu, 0.2$), $\mu \sim$ Normal$(1, 1)$

Figure 9: Structural comparison II. The predicted means are color-coded.

## D.7 Reduced-memory agents

In order to understand outcomes when the optimal policy is not in the search space we investigated the performance of a series of reduced-memory baselines. These were implemented with purely feedfoward architectures, which observed a context window of the previous $k$ timesteps (padded for $t < k$), rather than with an LSTM. Short context windows dramatically impaired performance, and the degree to which longer context windows allowed for improved performance was strongly task-dependent. In some cases (Dirichlet and high-precision Gaussian), extending the context window to match the episode length almost completely recovers performance, whereas in other cases performance plateaus.

(a) $\theta_1, \theta_2 \sim \mathrm{Beta}(1,1)$

(b) $\theta_1, \sim \mathrm{Beta}(2,1)$, $\theta_2 \sim \mathrm{Beta}(1,2)$

(c) $\mu_1, \mu_2 \sim \mathrm{Normal}(0,1)$

(d) $\mu_1, \mu_2 \sim \mathrm{Normal}(0,0.1)$

Figure 10: Structural comparison III (bandit tasks). Action probabilities are color-coded.

Figure 11: Convergence plots for our prediction tasks, showing 10 steps of generalisation (demarcated by grey dashed line).

(a) $\theta_1, \theta_2 \sim \text{Beta}(1, 1)$.

(b) $\theta_1 \sim \text{Beta}(2, 1)$, $\theta_2 \sim \text{Beta}(1, 2)$

(c) $\mu_1, \mu_2 \sim \mathcal{N}(0, 1)$

(d) $\mu_1, \mu_2 \sim \mathcal{N}(0, 0.01)$

Figure 12: Convergence plots for bandit tasks, showing 10 steps of generalisation (demarcated by grey dashed line).

(a) Prediction tasks.

(b) Bandit tasks

Figure 13: Performance as a percentage of LSTM agent score for reduced-memory baselines. Solid line is mean over 20 trials, shaded area shows standard error of the mean over 20 repetitions. Reduced-memory baselines are feedforward agents trained with a fixed-width context of past observations. Adjusting the context width scales the amount of history the agent can use when computing a prediction/action decision.

## Footnotes

[8]The reference trajectory is always generated from the fully trained RNN agent—also when analyzing RNN agents during training.