[Reviews · NeurIPS 2020]

Review 1

Summary and Contributions: Performs empirical tests of a previously stated theoretical observation that sets up a mapping between optimal Bayesian agents and RNN agents. (Most of the tasks are actually prediction tasks, so it's really Bayes filters and RNN filters.) By training a sufficiently large RNN with sufficiently many training examples and problem instances and the right training objective, it learns to mimic the Bayesian algorithm quite well.

Strengths: The value of this is in laying out the RNN-based and Bayesian processes side by side and in bringing this very simple (but important) idea to the attention of the metalearning community (which has IMHO mostly proceeded in blithe ignorance of what is going on). Most of that community views metalearning as "learning how to learn" but it's far more likely (as a Bayesian would say) to be just learning a better prior. The empirical results are solid and nicely illustrated, although for obvious reasons the paper considers only small, conjugate tasks.

Weaknesses: Perhaps the rhetoric of the paper is a little overheated, with lots of ITALICS and claims of novelty and significance that exceed the actual findings (see below). The basic ideas do not seem all that "striking". Yes, of course if we have a parameterized policy family that includes the optimal policy by design, and we train it with feedback such that the optimal policy is the one that maximizes the feedback, then it works. Note that the "target distribution" can be thought of as an initial stochastic step in a single (PO)MDP that samples the problem parameters, so the process is learning a policy for that POMDP. Where [10] (for which the authors are of course not necessarily responsible) says "Essentially, memory-based meta-learning translates the hard problem of probabilistic sequential inference into a regression problem," this is exactly what Monte Carlo RL does. See AIMA p 833 "This means that we have reduced reinforcement learning to a standard inductive learning problem, as discussed in Chapter 18." But AIMA goes on to say, "Direct utility estimation succeeds in reducing the reinforcement learning problem to an inductive learning problem, about which much is known. Unfortunately, it misses a very important source of information, namely, the fact that the utilities of states are not independent!" The implications for sample complexity are important. The number of training runs required to fit these very simple functions is eye-watering. If anything, this inefficiency is the most surprising result of the paper. There are also long-standing connections between RNN filtering and Bayes filtering (see below). Finally, the paper omits significant prior work within the metalearning community on formulating metalearning as Bayesian inference: "Recasting gradient-based metalearning as hierarchical Bayes" (Grant et al), which is the MAML authors' own Bayesian reinterpretation of MAML. Interestingly, this papers predates [49], which is someone else's proto-analysis of MAML.

Correctness: Generally yes, within the scope of the setup and experiments. The claims go beyond that, e.g., in the abstract, "Our results suggest that memory-based meta-learning is a general technique for numerically approximating Bayes-optimal agents—that is, even for task distributions for which we currently don’t possess tractable models." and l.262 "While most aspects of our methodology scale up well to more complex agents" Success on training a policy for a two-armed Bernoulli bandit (where the optimal value function is actually very smooth and simple) with 100 million trials does not provide evidence for these claims. Moreover, there is no guarantee that the mapping from the sufficient statistics to the decision is a simple one. In chess and Go, for example, it is not simple, which is why programs use lookahead. Your RNN agents cannot do that as formulated.

Clarity: The paper is generally very clear. l.70 One would expect that g be described, with Bayesian updating etc. Minor comments: Title: Meta-Trained -> Meta-trained l.72 i.e. -> i.e., [passim] l.76 "it is sufficient to remember the total sum of rewards and the number of pulls for each arm" - this is grammatically ambiguous and the natural reading [[the total sum of rewards] and [the number of pulls for each arm]] is wrong. Better might be "the number of successes and failures for each arm". Figure 1(b) Bernouli -> Bernoulli l.292 "in AI research, and development of powerful AI algorithms and applications" - drop comma, maybe "and in the" l.294 allow to separate -> allow one to separate References: Some inadequate references, e.g., "Citeseer, 1990". Many capitalization errors, e.g., lstm, ai, journal names, conference names, etc.

Relation to Prior Work: It seems the main ideas appeared already in reference [10], and the additional contribution of this paper is mostly the experiments and the simulation definitions (which are pretty obvious, and do not really require "inspiration from theoretical computer science"). There are several works not cited that draw parallels between Bayes filters and RNNs, e.g., Lim et al Recurrent Neural Filters: Learning Independent Bayesian Filtering Steps for Time Series Prediction Krishnan et al Structured Inference Networks for Nonlinear State Space Models There are also many other (easily retrieved) papers on mapping RNNs to specific Bayesian filters such as HMMs and Kalman filters. I would say also that the work of Rosenschein and Kaelbling on circuit implementations of goal-seeking agents exhibits the same core idea except in the deterministic context: a perfectly rational agent can be implemented as a register machine (the Boolean circuit analogue of an RNN).

Reproducibility: Yes

Additional Feedback: I think the paper could be improved by being more neutral about metalearning's miraculous properties and prospects and focusing on this Bayesian reinterpretation of what metalearning is doing. This might have more impact on changing the direction of metalearning research. Thanks for the very reasonable response to the reviews. I have raised my score and look forward to seeing the final paper. Indeed, I think it might be good for an oral plenary, given its pedagogical value for the broad ML community.


Review 2

Summary and Contributions: This paper empirically investigates the hypothesis that memory based meta-learning (ie. meta-learning by an RNN inducing adaptive task-specific policies) results in Bayes-optimal solutions. This is done by comparing metalearned systems against Bayes-optimal solutions on both prediction and reinforcement learning tasks. This comparison is done both in terms of the behavior of the systems and by attempting to map the state machines induced by both approaches to one another. The results show a clear correspondence, although also show that while the states of the metalearned representations can be mapped to the Bayes-optimal representations the reverse doesn't hold.

Strengths: This is an impressively detailed analysis of the solution discovered by memory-based meta-learning and makes a valuable contribution insofar as those methods remain relatively opaque. The analysis is motivated by theoretical claims, and takes a principled approach to evaluating those claims. In addition to the empirical results, the paper contributes a methodology for establishing correspondences between systems that have continuous states.

Weaknesses: 1. This might reflect my biases, but my main question was why we might expect these systems not to be Bayes-optimal on the relatively straightforward problems investigated in the paper. More work needs to be done at the start of the paper establishing the alternative hypotheses and gearing the analyses and results towards ruling those out. At present this is mostly a confirmatory analysis -- what needs to be pushed to really test that hypothesis? Laying out alternatives and the experiments that would discriminate between them seems critical. 2. To that end, I thought the inability to map the Bayes-optimal states to the metalearned states was particularly interesting. While this is minimized in the paper, it seems like the kind of thing that could reflect the networks having learned spurious correlations in the input data. Based on my reading, the evaluation of the systems was done on tasks that they had been trained on -- there wasn't an evaluation in a generalization context. My guess is that those differences in representation could result in meaningful differences in behavior in a generalization setting, in a way that could lead to the observed correspondence breaking down. I would like to see this analysis extended to (meta-)generalization across tasks to assess this.

Correctness: Yes

Clarity: Yes. One small point: in the introduction the text talks about comparing systems at the "computational level". This is used informally but could be spelled out or related to existing ideas (e.g., David Marr's notion of a computational level of analysis in his 1982 book "Vision").

Relation to Prior Work: Yes

Reproducibility: Yes

Additional Feedback:


Review 3

Summary and Contributions: This work is set to empirically validate a recent claim that memory-based meta-learning agents are Bayes-optimal. By focusing on simple synthetic tasks for which the optimal solution is known, the authors analyze RNN agents and show that the RNN agents are computationally and behaviorally similar to the Bayes-optimal agents when converged. For the behavioral analysis, the authors look at the KL-divergence and the absolute reward difference (in the case of bandit tasks) between the RNN and Bayes-optimal agents. For the structural and computational analysis, they check whether two agents can simulate one another. The simulation test requires discretization and compression of the RNN state or embedding of the Bayes-optimal agent state. To this end, mapping functions parameterized by a neural net were used.

Strengths: The paper addresses an important problem relating to interpretation and analysis of black box learners – RNN in this work. Analyzing a meta-learning method is even challenging due a meta-learned behavior. The authors made a sensible set of decisions to approach the problem. The work is well executed and scientifically sound. The findings from the behavioral and convergence analysis based on the dissimilarity measures can be valuable to the community.

Weaknesses: The structural analysis involves fitting an expressive function approximator (3-layer FF neural net in their experiment) for the mapping function. I am having a difficulty to tell whether their findings on the structural analysis due to this neural net. So, I am not fully convinced by this particular result. I think their experiment is missing an additional baseline. It makes sense to include a third agent that has no or limited memory – an agent that looks only at the last n observations (e.g. n=3). Showing that this kind of agent doesn’t exhibit the same behavior as the optimal solution by using the analysis taken could validate their approach. The related work section is missing. Specially, a broad related work discussion on neural net interpretability and memory-based meta-learning could have been helpful. The authors didn’t discuss why a specific approach was taken. For example, why simulation-based technique was used or what are the other alternatives that could have been used instead?

Correctness: I don’t see any major flaw in this work. Please see my comment above.

Clarity: The paper is overall well written and clear.

Relation to Prior Work: Please see my comment above.

Reproducibility: Yes

Additional Feedback: In figure 1, only first 30 steps are shown what happens if you go beyond? agents diverge? Cite the other memory-based meta-learning approaches (or explain in rebuttal why this is not appropriate): Munkhdalai et al., Meta networks. ICML 2017 Munkhdalai et al., Metalearned neural memory. Advances in Neural Information Processing Systems 2019 After Rebuttal --------------------------------------- My concerns were addressed during rebuttal and this is an interesting work that I would like to see more of. Therefore, I am increasing my score to 7.


Review 4

Summary and Contributions: Update: Based on the input from the other reviewers and the fact that one at least one of my issues were fully addressed in the rebuttal I will raise my score somewhat and recommend an accept. However, I would still like to see an SGD baseline added in the final paper if possible. The paper attempts to show that agents trained using memory-based meta learning behave approximately Bayes-optimally. This is accomplished by comparing meta trained agents to the optimal solution in a number of tasks where the optimal solution is tractable. Further, the paper introduces simulation based methods for comparing solutions that is used to empirically show the equivalence.

Strengths: The paper is polished and rigorous in most aspect and connect the empirical evidence to theory in a nice way. Further, the problem domain is of broad interest and bring some interesting new insights to the area.

Weaknesses: - My main objection to this work is the lack of baselines. In the structural comparison results they include a random baseline in the appendix which is great but I would be very interested to know how well meta-learning compares to, e.g. one or a few flavors of SGD. If nothing else so to get a better sense of the scale in the results. Maybe there is a good reason as to why such a baseline would not make any sense but if you agree I think that would strengthen the paper. - In section 4.3 you state that you use the meta-learner to generate the input streams that you use to compute the results. How does this impact the validity of the results? And does the results differ much if you instead use the "optimal" agent to generate the stream?

Correctness: The paper is rigorous in most aspect and as far I can see correct. My only concern would be that they claim that the "meta-learners behave virtually indistinguishable from Bayes-optimal agents" which I found as overselling the results a bit but since this is not a very exact statement I guess it is alright.

Clarity: The paper is well written and mostly easy to read. That said, personally I would prefer if the method/result sections was restructured to be less fragmented between experiments but other than that the text read fine.

Relation to Prior Work: Yes

Reproducibility: Yes

Additional Feedback: On line 192 I think it should be Appendix D.6

[Author Response · NeurIPS 2020]

**Overall comments** We thank the reviewers for their detailed and helpful comments. We followed the reviewers' suggestions and have significantly expanded our discussion of related work into a separate section, incorporating the references mentioned (R1, R3). As suggested by R1 and R4 we have changed our writing to a more neutral tone, and mention the claims flagged by R1 as directions to study more rigorously based on our findings, with appropriate caveats regarding convergence and inclusion of the optimal policy in the search space. We also performed two experiments investigating memory-limited agents (including no memory) as suggested by R3, and generalization (R2, R3).

(a)

(b)

(c)

Figure 1: **a, b**: Effect of reducing memory on prediction and bandits respectively **c**: Extended generalisation.

**Reduced-memory baselines (R3)** As suggested by R3, we added an additional baseline: an agent with reduced memory capacity. This agent was implemented using a feedforward network given the $n$ most recent timesteps. Results are shown in Figure 1A, B. These results also provide some information on outcomes when the Bayes-optimal policy is not in the set of policies - an interesting question suggested by the comments of R1 and R2.

**Further relevant work (R1, R3)** R1 mentions a number of papers on connections between RNNs and Bayes filtering, and highlights two papers which define new neural architectures for Bayes filtering. Unlike these papers, our manuscript focuses on the way in which Bayes filtering naturally arises from metalearning. This point—although clear once spelled out formally—is not widely appreciated in the metalearning community, as R1 notes (apart from the paper by Grant et al.). We also appreciate the pointers to Rosenschein and Kaelbling's work on situated/embedded automata. R3 highlights two related approaches to memory-based metalearning using fast and slow weights and a differentiable memory which we now include in an expanded related work section, as well as a section on interpretability.

**Clarification of simulation results (R3)** R3 refers to the expressivity of the map between the latent states of the Bayes-optimal and RNN agents as a possible weakness in our analysis. This map is regularised by a PCA of the RNN agent's latent state before the MLP. Importantly, a failure of simulation is a failure of injectivity: if a single state in one agent must be mapped to two distinct states in another then simulation fails, and no amount of expressivity in the map between states will save it. This occurs when two trajectories lead to the same state in one agent but not another (for instance if exchangeability has not been fully learned). We have clarified this in our discussion.

**Is Bayes-optimality a surprising result? (R1, R2)** R1 and R2 point out that our main findings are not completely surprising. While the behavioral correspondence is expected, it is theoretically unclear how (fully trained) RNN meta-learners internally organize sufficient stats. The fact that our particular structural analysis method is successful suggests a smooth and fairly structured organization of internal states in a (quasi-?) Euclidean space (interestingly similar organizational patterns have also been found in biological neural networks: doi 10.1101/461129). This, in turn, might also play an interesting role in a theoretical understanding of generalization (see next point).

**Limitations of claims and Bayes-optimality in the generalization regime (R2, R1, R3)** Our claims regard trained RNN meta-learners "on-distribution". We are currently not aware of theoretical predictions for generalization in meta-learners. We report positive results for evaluation on extended episodes in Fig. 1 in the paper, and we have also run experiments testing generalisation substantially beyond the 20 timestep horizon during training, by evaluating up to 1,000 timesteps (Figure 1C above, not cherry-picked). Meta-learned solutions generalise surprisingly well, suggesting that a generalisable update rule has been learned, but the mechanisms underlying this are not understood empirically or theoretically. In the absence of an adequate explanation, we would prefer to leave these mechanisms to further study.

**Other comments: R1**: Sample-inefficiency of meta-learning as a regression problem. Thanks for highlighting the connection to MC RL and the discussion in AIMA. We agree with the core issue and note that sample-efficiency was not a primary concern in our study - though it is a very active area of research: arXiv 2006.16507 and 2006.05094. Although Figures 11 and 12 show convergence 1-3 orders of magnitude before training stops, this is still sample-inefficient.
**R2**: comparison at the "computational level" —we did intend this in the Marrian sense, and now cite appropriately.
**R2**: Alternative hypotheses. We agree that these are interesting points to follow up on. However, if meta-training converges properly (regardless of problem complexity) there is no behavioral deviation from Bayes-optimality and the theory says little about the computational structure. Before convergence or off meta-distribution, a theoretical understanding is currently lacking.
**R4**: using the Bayes-optimal agent to generate inputs leads to small numerical changes, but no change to the conclusions.
**R4**: SGD baseline. We assume this refers to learning by SGD on a single sample from the environment prior. We were unable to implement this in time for the rebuttals, but do not expect that such a solution would generalise.

[Meta-Review · NeurIPS 2020]

The reviewers are all positive on this paper. However, I am skeptical about the significance of results on the synthetic datasets considered. I understand that the point is to compare Bayes-optimal performance with RNN performance and hence one must be able to determine Bayes-optimal performance. But these analytically solvable tasks seem fundamentally different from problems of real significance. Hence the acceptance recommendation.